# Simulating Society Requires Simulating Thought

**Chance Jiajie Li[1]\*, Jiayi Wu[9]\*, Zhenze Mo[8], Ao Qu[4], Yuhan Tang[5],**
**Kaiya Ivy Zhao[2,3], Yulu Gan[2], Jie Fan[2,7]†, Jiangbo Yu[10],**
**Jinhua Zhao[4,5,6], Paul Pu Liang[1,2], Luis Alonso[1], Kent Larson[1]**

[1]MIT Media Lab    [2]MIT EECS    [3]MIT BCS    [4]MIT IDSS    [5]MIT CEE    [6]MIT DUSP
[7]MIT Architecture    [8]Northeastern University    [9]Brown University    [10]McGill University

\*Equal contribution.

†Now at Google.

Correspondence to: `jiajie@mit.edu`

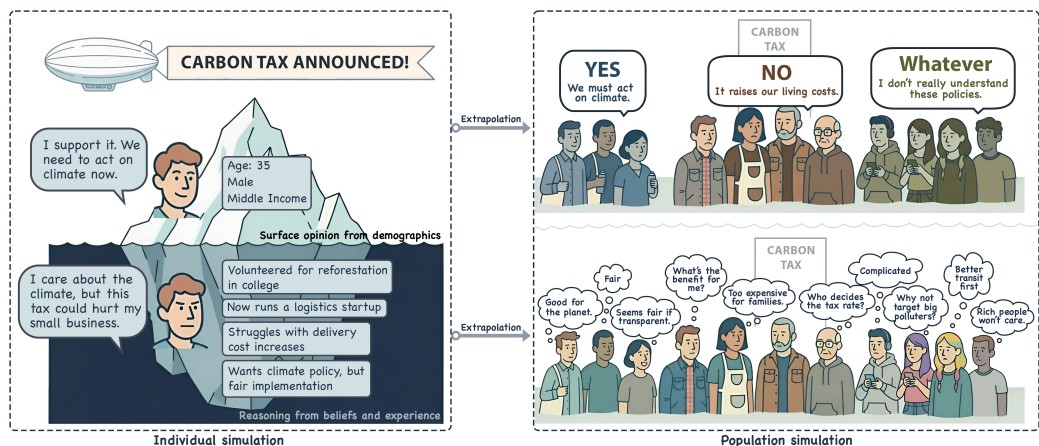

Figure 1: **From surface imitation to cognitively grounded social simulation.** Current LLM-based simulations (top left) capture only *surface opinions*, shaped by demographics or language patterns, while the deeper *belief formation processes* remain unmodeled (bottom left, beneath the waterline). This yields population-level simulations that are *flattened and stereotyped*, reflecting aggregated personas rather than genuine diversity. In contrast, cognitively grounded reasoning (bottom right) models the latent belief dynamics behind individual decisions, producing collective patterns that are heterogeneous, interpretable, and causally faithful.

## Abstract

**Simulating society with large language models (LLMs), we argue, requires more than generating plausible behavior; it demands cognitively grounded reasoning that is structured, revisable, and traceable.** LLM-based agents are increasingly used to simulate individual and group behavior, primarily through prompting and supervised fine-tuning. Yet current simulations remain grounded in a behaviorist *"demographics in, behavior out"* paradigm, focusing on surface-level plausibility. As a result, they often lack internal coherence, causal reasoning, and belief traceability, which makes them unreliable for modeling how people reason, deliberate, and respond to interventions.

To address this, we present a **conceptual modeling paradigm**, **Generative Minds (GenMinds)**, which draws from cognitive science to support structured belief representations in generative agents. To evaluate such agents, we introduce the **RE-CAP** (*REconstructing CAusal Paths*) framework, a benchmark designed to assess reasoning fidelity via causal traceability, demographic grounding, and interven-

tion consistency. These contributions advance a broader shift: from surface-level mimicry to generative agents that simulate thought—not just language—for social simulations.

# 1   Introduction

**The Rise of LLMs in Social Simulation.**   Over the past two years, LLMs have increasingly become dominant tools for simulating human behavior across language [1, 2], vision [3] and decision-making domains [4].In the field of social simulation, LLMs are now commonly used to emulate public opinions, stakeholder interactions, and policy responses under diverse scenarios [5, 6, 7].

**An Oversimplified Paradigm: Demographics In, Behavior Out.**   Despite the growing use of LLMs in social simulation, most current models rely on simplified input-output mappings, producing behavior based on surface cues rather than simulating the internal belief dynamics behind decisions. This approach mirrors the logic of *behaviorism* in psychology, which models behavior as a function of external stimuli while ignoring internal cognitive states. The limitations of this paradigm echo a broader historical tension between *behaviorism*, *cognitivism*, and *constructivism*[8, 9]: while *cognitivism* emphasized structured internal representations and causal reasoning, and *constructivism* further argued that beliefs are continually shaped by individual and social experience, existing LLM-based agents remain far from either. They typically exhibit shallow reasoning, frequent hallucinations, and limited understanding of causal and contextual dynamics in socially-salient domains such as upzoning, surveillance, or healthcare access, precisely the domains where reasoning fidelity matters most [10, 11].

$$Behaviorism \longrightarrow Cognitivism \longrightarrow Constructivism$$

**Structural Failures: Modeling, Evaluation, and Calibration**   These failures stem directly from the behaviorist paradigm outlined above. By focusing on surface behavior instead of the reasoning behind it, most LLM-based simulations face fundamental limitations in both modeling and evaluation.

In modeling, agents often rely on shallow input-output patterns, without representing how beliefs are formed, updated, or justified. As a result, their internal reasoning is difficult to inspect, especially when context changes. It is hard to determine how a new policy or scenario influences an agent's judgment, or why a particular decision is made. Without access to reasoning traces, agents cannot support diagnostic explanation, causal attribution, or meaningful intervention — all of which are critical for multi-stakeholder policy simulations. Even when models succeed at surface-level generation, they are difficult to adapt to new domains. Fine-tuning LLMs for specific contexts often requires significant compute and high-quality datasets, which are rarely available in real-world policy settings. Yet effective simulation depends on exactly the opposite: the ability to represent evolving stakeholder reasoning grounded in timely, localized information [12, 13].

In evaluation, models are typically judged by output plausibility or alignment with population-level trends[14], but such metrics say little about whether their reasoning is accurate, flexible, or aligned with how people actually think. Post-hoc output analysis is common, but it cannot substitute for reasoning-level evaluation. Aligning agents with real-world stakeholders requires individual-level data and internal benchmarks for reasoning fidelity, both of which are largely missing today.

**Toward Mechanistic and Individual-Level Alignment**   These limitations call for a shift in how we conceptualize generative social simulation—not as behavior mimicry, but as cognitive modeling. This paper takes up that call. Specifically, we propose leveraging ideas from Theory of Mind (ToM) and cognitive science to extract and simulate reusable, executable reasoning units—what we term **reasoning traces**—rather than simply mimicking human tone or persona [15, 16].

Unlike prompt-driven persona or character approaches that generate "average" group behaviors [17], cognitive models allow agents to represent beliefs, values, and causal assumptions in a compositional manner. This makes it possible to generalize to unseen scenarios, so long as the individual components of the reasoning trace are known. For example, if a stakeholder has previously reasoned about "density" and "transit," then when asked about a novel "transit-oriented development" policy, the agent can reuse those motifs to simulate beliefs without re-training.

Such compositionality is a cornerstone of human cognition, where reasoning emerges from fragments that are reusable, revisable, and structured across contexts [18, 19]. It also improves simulation fidelity: interactions between agents, or between agents and dynamic environments, can be represented through composable and transparent reasoning structures, enabling structured simulation at both micro and macro levels [20]. Moreover, reusing compositional and modular reasoning units reduces the need to

regenerate full-context reasoning at every step, improving both interpretability and computational efficiency.

**Position and Vision** **In this paper, we advocate moving beyond output-level alignment toward aligning the internal reasoning traces of generative agents.** Capturing the causal, compositional, and revisable structure of belief formation, which we refer to as reasoning fidelity, is essential for building cognitively faithful agents that simulate not only what people say but also how they think.

To support this argument, we:

- Illustrate how current approaches fall short by producing outputs that appear coherent but lack internal consistency, adaptability, or traceability;
- Theorize reasoning fidelity as a structural alignment problem grounded in cognitive science;
- Introduce a symbolic-neural framework for simulating belief formation through modular reasoning motifs and causal graphs;
- Present a methodology for extracting and simulating belief structures from natural language, enabling interpretability, counterfactual reasoning, and domain transfer.

In summary, this position paper argues that simulating human society requires more than generating plausible conversations. It requires **simulating the structure of human reasoning**. By grounding agents in modular belief representations and evaluating them on reasoning fidelity, we take a critical step toward building generative minds, not just generative outputs.

## 2 Social Simulation: Opportunities and Gaps

*Social simulation* has emerged as a high-impact use case for LLMs [6]. Traditionally, social science relies on surveys, experiments, fieldwork, or game-theoretic models to understand individual and group behavior [21]. While effective, these methods are expensive, hard to scale, and often face ethical and logistical challenges. The rise of LLM-driven agents offers a promising alternative capable of simulating human responses across a wide range of scenarios, roles, and interventions. Therefore, LLM-agent-based social simulation has been increasingly implemented in domains including policy modeling [22], behavior forecasting [23], annotation tasks [24], and opinion surveys[25].

Recent studies demonstrate that LLMs can emulate key aspects of human reasoning and decision-making [26, 27, 28, 29], enabling agents that perceive their environment, make context-sensitive decisions, and articulate motivations. When equipped with role-based prompting or persona conditioning [30, 31], these agents exhibit a property known as *algorithmic fidelity*—the ability to simulate how specific individuals or subgroups might respond in a given situation [25, 32].

Beyond single-agent settings, multi-agent simulations extend this potential by modeling the interactions, conflicts, and consensus dynamics among synthetic populations. However, faithfully simulating social processes introduces several critical requirements:

1. **Fidelity**: Simulated agents should be faithful in respect to individual human reasoning [33, 34].
2. **Individuality**: Simulations should preserve individual-level heterogeneity [33, 35], capturing the positional and contextual diversity of human reasoning.
3. **Extrapolation**: A core challenge is out-of-distribution generalization. Agents must reason in novel scenarios and generalize from skewed population subsets to broader groups [34, 35], requiring belief updates and counterfactual reasoning under uncertainty.

While many recent works aim to create more "human-like" agents, few clearly define what this term means, or how such fidelity is to be evaluated. As a result, although large-scale agent environments now enable rapid, low-cost exploration of collective behavior, due to the lack of internal transparency and explainability of language models today [36], most simulations remain behaviorist at their core, thus simulated societies risk reflecting the biases and homogeneity of the prompts rather than the heterogeneity of human internal belief structures. These methodological and epistemic gaps motivate a shift toward cognitively grounded simulation.

## 3 Problem Statement: Beyond Behavioral Plausibility in Generative Social Simulation

Most existing efforts to align LLM-based agents focus on model *behavior* [29]: Do agents take stances, express preferences, or engage in natural-sounding conversations as human subjects they aim to imitate? This behavior-centric view is reinforced by popular techniques such as reinforcement learning from human feedback (RLHF) [37, 38], persona prompting [39], and chain-of-thought (CoT)

generation [26, 40]. These methods optimize for behavioral *plausibility* rather than structural fidelity of human reasoning.

The behavioral-centric framework overlooks a critical problem: output plausibility is not equivalent to cognitive alignment. In this section, we argue that behavioral fluency fails to serve as a proxy of agents' reasoning fidelity. Without structural representations of belief, sensitivity to counterfactual intervention, or positional diversity of individual human subjects, generative agents risk producing surface-level plausible outputs that are epistemically unaligned. We identify two core challenges of the current agent simulation paradigm: (1) **fidelity**, the agent's capacity for coherent, revisable, and causally grounded belief formation and (2) **individuality**, the agent's ability to model distributed and positional human reasoning, and show how both remain underspecified in existing evaluation **metrics**.

## 3.1 Fidelity: Coherence, Traceability, and Causal Grounding

One crucial aspect of social simulation is that it must ensure agents think in ways that are causally structured, internally coherent, and dynamically revisable. Current LLM-based agents fail to meet these standards, and this mismatch between surface-level plausibility and internal structural fidelity manifests in a set of persisting reasoning failures.

### 3.1.1 Traceability and Interpretability

Current LLM-based agents presents two essential gaps in reasoning fidelity: (1) *decoding faithfulness mismatch*, where generated reasoning traces in agent's output generated traces diverge from the model's internal computational path; (2) *cognitive-alignment mismatch*, where the modeled inference path diverges from human belief formation. Subsequently, LLM-based agents face *intervention-invariance mismatch*, where belief updates fail under counterfactual perturbations.

The first critique is largely similar to those observed in CoT-related discussions [41, 42]: agents may produce fluent rationalizations, while their "reasoning traces" are constructed post hoc: assembled from language patterns rather than derived from an underlying belief model [43, 44]. Recent controlled studies confirm that the existing framework of LLM reasoning is largely a data mirage: Zhao et al. show that CoT performance collapses when tasks, the length of reasoning chain, or prompt format deviate moderately from the model's training-set distribution [45]. This finding reinforces that current CoT outputs are not a faithful representation of the model's actual reasoning processes, as they lack stability under distributional shift and therefore fail to reveal the causal decision paths or the underlying dependency assumptions guiding them. While there are some promising approaches beginning to incorporate structured representations, such as knowledge graphs, belief graphs, and additional reasoning layers [46, 47, 48], most of them are domain-specific and disconnected from live generative processes; also, none have yet been operationalized within interactive, generative social simulation pipelines.

The second critique concerns the presumed alignment between model reasoning and human reasoning, which remains largely untested. Although emerging proposals in cognitive science examines LLMs through theory-of-mind and belief-attribution paradigms [49], these studies largely evaluate behavioral correlates (e.g., predicting others' beliefs or heuristics) rather than mapping internal reasoning operators to human causal schemas. Moreover, no standardized benchmark operationalizes measurable correspondences between model-internal belief transitions and human causal inference sequences. Therefore, claims of "human-like reasoning" remain more speculative than empirically grounded, lacking any quantitative measure of cognitive fidelity.

**Alternative View: Post-hoc rationalization may be cognitively authentic and functionally sufficient.** Drawing from work in social and cognitive psychology, one might argue that people often rationalize decisions or beliefs after the fact [50, 51]. LLMs' tendency to construct rationales retroactively might therefore be seen not as a defect, but as a cognitive parallel to human behaviors.
**Response.** We do not target post-hoc rationalization per se, but its total detachment from structured belief representation. Human justifications are often imperfect but still rely on internal models of causality, memory, and values and are traceable thereafter [52], whereas LLMs produce rationalizations without structured anchoring. *Form* (i.e. the shape of the rationale) is not the same as *function* (i.e. the structural, belief-guided deliberation). Agents must operate on explicit, traceable structures to simulate human reasoning.

### 3.1.2 Counterfactual Intervention Sensitivity and Belief Revision

In social simulations, agents are expected to revise their stances when key assumptions or contextual conditions change, which is a hallmark of human reasoning known as counterfactual intervention

sensitivity [53, 54]. However, due to the lack of an internal causal structure that anchors beliefs to causes and consequences [11, 55], current LLMs and LLM-based generative agents often respond to such interventions with inertia or token-level paraphrasing. As a result, they cannot explain why a particular belief might hold under some conditions but not others, nor can they simulate the effects of counterfactual changes.

This structural deficiency manifests as inconsistency across prompts or dialogue turns: empirical studies have shown that LLM-based agents may support one policy in one scenario, then oppose in another without any causal reasoning-trace grounding [56, 57]. While there have been research efforts in grounding models and model-based agents with causal memory [58] and knowledge graph [59], most of them focus on graph discovery and construction [60, 61] in specific knowledge domains rather than general human belief systems. Without an explicit model of how beliefs are formed, revised, and connected, agents' utterances are generated in isolation—locally plausible, but globally incoherent.

**Alternative View: Human cognition is non-monotonic and contextually fluid, thus demanding coherence is unrealistic.** One might argue that humans often hold incoherent or even contradictory beliefs. Demanding that agents simulate perfectly consistent beliefs risks idealizing cognition and misrepresenting actual human messiness [62, 63].

**Response.** We do not call for rigid logical coherence or monotonic reasoning. Rather, our claim is modest: agents should be able to faithfully simulate belief revision under counterfactual assumptions—not that they maintain perfect consistency across all scenarios. Furthermore, human belief systems can be messy and involve incoherent or even contradictory stances, but it doesn't mean they're structureless. In human cognition, contradictions are often meaningful, reflecting a set of ambivalent conventions and priors in the social system at large [64]; in contrast, LLMs generate contradictions without memory, deliberation, or causal record. We don't require logical perfection but rather grounded incoherence, that agents should be able to simulate how humans arrive at contradictory views and under what conditions those contradictions persist or resolve.

Taken together, these observations define reasoning fidelity as the preservation of structured belief dynamics under decoding transparency, cognitive alignment, and counterfactual intervention sensitivity. Current autoregressive architectures optimize next-token likelihood rather than belief-state transitions, making fidelity an architectural limitation rather than a prompting issue [65]. Until this gap is addressed, generative agents will continue to exhibit the symptoms of alignment while remaining fundamentally unaligned at the level of thought.

### 3.2 Individuality: Heterogeneity and Positional Reasoning in Social Simulation

Alongside reasoning fidelity, social simulation requires an additional layer of alignment: positional individuality. In particular, we define individuality as the preservation of heterogeneity in agents' latent belief and value representations under shared generative priors. Current LLM-based agents, optimized under shared autoregressive parameters and global priors, collapse toward the mean of the pretraining distribution, erasing structured heterogeneity.

When deployed in real-world contexts, such as civic simulations [66, 67], participatory policy design [68, 69], stakeholder modeling [70], agents that lack internal reasoning fidelity may produce outputs that appear thoughtful, yet encode no coherent decision process beneath the text. This disconnect introduces a set of critical downstream failures:

#### 3.2.1 Illusion of Consensus in Multi-Agent Systems

One critical downstream failure of LLM-based agent social simulation is the illusion of consensus. Recent studies demonstrate that LLMs in multi-agent setups exhibit conformity behavior, and the benchmark shows virtually all models converge in behavior under majority-pressure protocols [71].

This convergence reflects a deeper statistical mechanism: models trained to minimize token-level loss implicitly learn to average across pre-training data distributions, biasing their conditional likelihoods toward high-frequency, socially moderate continuations [72, 73]. When multiple model-based agents interact, this produces a form of *synthetic agreement* aligned with a median perspective that masks underlying conflict, complexity, and epistemic independence [74, 75]. In multi-agent policy or deliberation simulations, this can yield systematically misleading inferences: agents appear to "agree" on a position not because of shared reasoning, but because their generative priors and cross-conditioning push them toward a median narrative that suppresses disagreement and complexity.

### 3.2.2 Flattened Outputs in Demographic Conditioning

A related but more insidious form of convergence occurs within demographic conditioning. Since LLMs are trained on aggregated corpora and lack explicit conditioning on intersecting social variables, their generative prior implicitly factorizes across dimensions such as race, class,a nd gender. This yields identity flattening, that agents reproduce majority-class correlations that dominate pretraining statistics, producing homogenous or stereotypical portrayals of social groups [76, 77], with resulting responses minimizing token-level loss but erasing intersectional variation. This leads to epistemic harm: the rich, positional knowledge of real-world stakeholders is replaced with monolithic, decontextualized simulations.

**Alternative View: Generalization over identity categories is necessary for tractable simulation.** One might argue that abstractions over demographic identities are unavoidable and, in fact, desirable when building simulators at scale considering model tractability [78]. Identity flattening may be viewed as a form of necessary regularization.
**Response.** Our critique is not about the use of abstraction per se, but the fact that LLMs abstract without modeling the joint distribution of beliefs, values, and positionality conditioned on intersecting variables (e.g., age × race × class × institutional exposure) and how abstraction, subsequently, is operationalized without epistemic grounding [76]. In social simulations, such abstractions introduce bias, undermine group heterogeneity, and lack epistemic representativeness, especially when the simulation output is used to inform policy or governance decisions [79].

As agents are increasingly used to test policy options, simulate deliberative processes, or represent groups in synthetic social systems, these reasoning deficiencies risk being institutionalized. Decision-makers may take model outputs at face value, unaware that these outputs do not derive from any concrete belief structure. Ultimately, when agents simulate without reasoning, the outputs they generate can erode trust, misinform policy, and flatten the epistemic diversity of the very populations they are meant to represent [76, 80, 81, 82].

## 3.3 Evaluation Gaps: Structural Limitations of Current Benchmarks

The deficiencies in reasoning fidelity and individuality are compounded by a third layer: evaluation misalignment. Despite the growing sophistication of generative agents, current benchmarks remain optimized for stylistic fluency, local coherence, and plausibility of individual model behavior and output. Most benchmarks treat language as a proxy of thought, implicitly assuming that coherent expression entails coherent reasoning. As a result, current evaluations risk rewarding surface-level coherence while overlooking causal consistency and belief heterogeneity [71]. This first creates what we term as *traceability gap*: benchmarks assess outputs instead of reasoning trajectories. Stance classification tasks, for example, check whether a model picks a side but remain agnostic about how or why that stance was formed [83, 84]. Dialogue benchmarks reward conversational smoothness, even when agents flip positions over time [85, 86, 87].

The second limitation concerns *intervention blindness*: most benchmarks assess agents on static inputs but fail to measure their belief revision under counterfactual interventions or other hypothetical perturbations [88, 89, 90, 91, 92]. Finally, current benchmarks leave agents' positional individuality largely unaddressed. Evaluation suites typically aggregates performance across stances, computing mean accuracy or sentiment agreement but rarely quantify inter-agent divergence or distributional variance. Recent work on pluralistic alignment has begun moving in this direction, measuring whether models maintain epistemic diversity or support multiple internally coherent responses [93, 94, 95]; yet these methods remain limited to small-scale reasoning or dialogue tasks and have not been adopted within social simulation pipelines, where evaluation still focuses on output fluency and stance accuracy.

In sum, evaluation protocols for generative agents remain behaviorally calibrated but structurally ungrounded. Bridging this gap requires a new generation of benchmarks that treat reasoning as a structured process rather than a stylistic performance.

## 4 What Does Human-Like Reasoning Entail?

The failures outlined above stem from a core mismatch between behavioral alignment and structural reasoning alignment. This section turns from diagnosis to design: what structural properties must agents possess to simulate human-like reasoning? We outline potential modeling paradigm shifts in social simulation, theoretical foundations drawn from cognitive science, and definitions of reasoning fidelity.

## 4.1 Modeling Paradigms in Social Simulation: A Cognitive Turn

While recent efforts in generative agent research focus on improving behavioral plausibility through techniques like persona prompting [30, 31], reinforcement learning from human feedback (RLHF) [37, 38], and chain-of-thought (CoT) generation [26, 96, 42], these methods share a common assumption: that plausible language implies plausible reasoning.

Our position challenges this assumption. These methods remain fundamentally *output-centric*, optimizing for stylistic fluency or stance alignment without simulating how beliefs are causally formed or revised. This often leads to post-hoc rationalizations, identity flattening, and the illusion of consensus.

By contrast, we propose a *cognition-centric* paradigm shift: modeling thought as a structured, revisable, and compositional process. Table 1 outlines this distinction.

| Dimension | Existing Paradigm | Our Proposal (GenMinds) |
|---|---|---|
| Reasoning Format | Token-level generation, post-hoc | Structured belief graphs, motifs |
| Belief Dynamics | Static or reset each prompt | Revisable via causal updates |
| Evaluation Lens | Output fluency, stance labels | Reasoning fidelity and adaptability |
| Social Representation | Averaged, flattened views | Divergent, positional cognition |

Table 1: Paradigm shift from output mimicry to cognitive modeling in generative agents.

## 4.2 Theoretical Foundations: Causal, Compositional, Revisable

To move beyond behavioral alignment, we must first define what it means to reason like a human.

Cognitive science offers a well-established answer. Decades of research suggest that human reasoning is not merely reactive output generation, but a process grounded in structured representations, counterfactual simulation, and dynamic belief updating [16, 18, 97]. From these foundations, we identify three defining features of human-like reasoning:

1. **Causal:** Humans reason in terms of causes and consequences. Even young children exhibit Bayesian-like inference over causal relationships and use interventions to test hypotheses about the world [16, 98]. Mental models are structured around "what caused what," emphasizing explanation rather than mere correlation. This causal orientation allows for robust generalization and counterfactual reasoning [97].
2. **Compositional:** Human reasoning is modular and reusable. Cognitive architectures operate by composing shared schemas—what we term *cognitive motifs*—that generalize across domains [18, 20]. These motifs support efficient reasoning by enabling agents to simulate belief structures without re-learning from scratch [19].
3. **Revisable:** Human beliefs evolve dynamically. When presented with new information or contradiction, individuals revise their prior assumptions. This capacity for belief updating has been modeled through probabilistic programming and counterfactual simulation frameworks [19, 99], capturing the adaptive, non-monotonic nature of human thought.

Taken together, the three dimensions of *causal*, *compositional*, and *revisable* reasoning form the foundation of what we call **reasoning fidelity**, defined as the structural integrity of belief formation and revision processes in generative agents.

## 4.3 Defining Reasoning Fidelity

We define **reasoning fidelity** as an agent's ability to construct, simulate, and revise a structured trace of belief formation that mirrors human causal reasoning patterns. This concept extends the dual-process model proposed by [99], in which language models interact with structured reasoning systems to model inference, belief, and decision-making.

Reasoning fidelity comprises three measurable properties:

1. **Traceability** — the ability to inspect how a belief or stance was formed through intermediate reasoning steps [100, 101];
2. **Counterfactual adaptability** — the capacity to revise beliefs predictably in response to interventions or changes in context [102, 103];
3. **Motif compositionality** — the reuse of modular causal structures (motifs) across different scenarios or domains [99, 104].

These properties define the core evaluation axes in the proposed **RECAP paradigm**, which shifts benchmarking from output plausibility to structural reasoning fidelity (Section 5). For example, traceability is assessed via motif-to-stance inference accuracy, adaptability through belief revision under hypothetical scenarios, and compositionality via motif reuse across unrelated topics.

This framework can be formed through explicit causal belief graphs, as illustrated in our proposed *GenMinds* architecture (Section 5). In such graphs, nodes represent causally relevant concepts (e.g., policy tradeoffs, values, or outcomes), and directed edges encode influence relationships. These graphs are derived from natural language using LLM-guided parsing and persist across interactions, enabling intervention analysis and reasoning trace reconstruction.

Importantly, this architecture is not tied to any particular implementation. While LLMs may serve as one plausible interface for extracting cognitive motifs, the core modeling contribution lies in structuring reasoning as revisable causal graphs. This approach is compatible with both symbolic and neural systems [99], and GenMinds exemplifies one such instantiation of this broader modeling principle.

At the evaluation level, reasoning fidelity fulfills emerging demands for cognitively grounded AI benchmarks [88]. It offers a testable, interpretable standard for assessing agent behavior that goes beyond language mimicry.

Yet current LLM-based agents fall short of this standard. Most optimize for surface alignment by producing plausible stances such as "I support policy X," without modeling the underlying belief process. They lack persistent belief states, causal coherence, and principled revision under counterfactuals. This results in brittle or contradictory responses, agreement bias between agents, and an absence of traceable justification.

# 5 Toward Cognitively Grounded Simulation: Modeling and Evaluation Principles

After outlining the cognitive foundations necessary for human-like reasoning, we translate these principles into a modeling and evaluation framework for cognitively grounded simulation. This includes *GenMinds*, which models structured belief formation, and *RECAP*, which evaluates reasoning fidelity in generative agents.

## 5.1 GenMinds: A Framework for Modeling Human-Like Reasoning

**Structured Thought Capture: From Semi-Structured Interviews to Causal Graphs.** To build generative agents that simulate human reasoning rather than merely output plausible stances, we propose modeling individuals' internal logic through **semi-structured interviews**, adaptively conducted by large language models (LLMs). These interviews elicit causal explanations in everyday language (e.g., "why do you support X?" "what does Y influence?"), which are then parsed into directed acyclic graphs representing the participant's belief structure [3]. Each node encodes a concept (e.g., fairness, safety, family needs), and each edge encodes a directional causal relation, with confidence and polarity scores.

**Shared Knowledge.** We introduce *cognitive motifs* as minimal causal reasoning units extracted from natural language. These motifs—e.g., "Surveillance → Crime Rate → Public Safety"—capture widely shared conceptual dependencies across individuals. When aggregated across interviews, they form a topology of commonly held belief structures.

We represent these motifs in a symbolic causal graph (CBN), enabling alignment of diverse opinions while maintaining transparency of reasoning. By grounding this structure in semi-structured interviews, we connect population-level reasoning to individual narratives.

**Inference via Symbolic–Neural Hybrid Graph Simulation.** We define reasoning as a form of forward inference over belief graphs: given a causal structure and an intervention (e.g., "increasing housing near transit"), the agent uses probabilistic updates (e.g., do-calculus) to simulate belief shifts and final stances. A language model selects relevant interventions and assembles motifs into a causal Bayesian network. This hybrid method ensures both interpretability and expressive power, enabling agents to trace "why" a conclusion was reached and what would change it.

**Be Aware of Unknown.** While causal motifs help model explicit reasoning patterns, real-world beliefs are often incomplete or contradictory. Our framework is designed to highlight missing links or uncertain dependencies by visualizing weakly supported or isolated nodes in the graph.

We encourage future systems to maintain uncertainty visualization and prompt-based elicitation to expand motif coverage, rather than overfitting to known paths. This allows belief modeling to remain adaptive and open-ended, rather than overly deterministic.

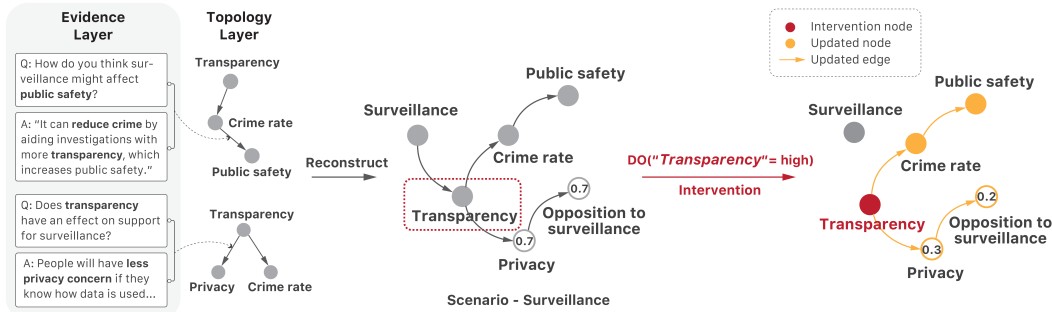

Figure 2: **Motif-based belief graph and intervention.** Natural language responses are parsed into motif-level causal links, forming a personalized belief graph. A simulated intervention on `Transparency` propagates downstream updates, shown as highlighted nodes and edges.

**Illustrative Example: From Interviews to Reasoning Agent Structures**

To concretize how motif-based causal reasoning operates in our framework, we present a real scenario from our semi-structured interviews on urban surveillance.

**Step 1: Extracting causal motifs from QA responses.** We start with Q and A responses annotated with concept nodes and directional relations. For instance:

- **QA#1:** *Q: How do you think surveillance might affect **public safety**? A: "It can **reduce crime** by aiding investigations with more **transparency**, which increases public safety."* ⇒ Motif: `Transparency → Crime rate → Public safety`
- **QA#2:** *Q: Does **transparency** have an effect on support for surveillance? A: "People will have **less privacy concern** if they know how data is used..."* ⇒ Motif: `Privacy ← Transparency → Crime rate`

**Step 2: Composing a Causal Belief Network.** These motifs are compiled into a belief graph representing the participant's reasoning. Nodes are concepts; edges indicate directional influence. Confidence scores are derived from motif density or respondent emphasis.

**Step 3: Simulating belief change via intervention.** We apply a hypothetical intervention:

$$\text{do (Transparency = high)}$$

This reflects a policy shift such as increasing camera accountability. Using belief propagation over the CBN, the downstream posteriors update as follows:

$$P(\text{Privacy Concern}) : 0.7 \rightarrow 0.3$$
$$P(\text{Opposition to Surveillance}) : 0.7 \rightarrow 0.2$$

This chain demonstrates the potential of motif-based causal modeling to simulate how real individuals update their beliefs in response to policy changes, thereby moving beyond static opinion snapshots.

## 5.2 RECAP: Principles for Evaluating Reasoning Fidelity

To advance cognitively aligned simulation, we propose a benchmark framework called **RECAP**, that shifts evaluation from surface-level correctness to the internal structure and coherence of reasoning.

**Design Principles.**

- **Traceability:** Can the agent construct a transparent chain of intermediate beliefs?
- **Demographic Sensitivity:** Can it represent diverse reasoning paths across identities or contexts?

- **Intervention Coherence:** Does it revise beliefs in response to hypothetical changes in a consistent, causally grounded way?

**Structure and Inputs.**
- Situated prompt in a morally or socially complex domain;
- Human-annotated responses capturing causal motifs and belief chains;
- A task such as graph reconstruction, stance explanation, or counterfactual reasoning that requires structured inference.

**Metrics.**
- *Motif Alignment:* Structural similarity between human and model belief graphs;
- *Belief Coherence:* Internal consistency of the model's reasoning trace;
- *Counterfactual Robustness:* Sensible belief updates under interventions.

**Grounding in Human Reasoning.** All items originate from real-world, semi-structured interviews, capturing how people explain and revise their beliefs. This grounding ensures the benchmark reflects the complexity and causal depth of actual human reasoning.

**Toward a Shared Format.** RECAP is not a static dataset but a replicable schema for structured reasoning evaluation. Grounded in human-derived motifs, it aims to promote interpretability, adaptability, and socially responsible agent design.

# 6 A Call for Cognitively Grounded Simulation

As large language models become embedded in social simulations and policy tools, we face a pivotal choice: whether to pursue agents that merely sound human, or agents that can reason in structured, human-like ways. This paper argues for the latter. We call for a shift from behavior-level mimicry to cognitively grounded reasoning, where agents represent beliefs, simulate causal relationships, revise assumptions, and reveal their internal logic.

We introduced *Generative Minds* and *RECAP* as conceptual scaffolds to support this shift—prioritizing reasoning fidelity, traceability, and epistemic diversity over surface plausibility. These are not fixed systems, but a framework for developing agents that simulate how people think, not just what they say.

This paradigm enables more transparent diagnostics, pluralistic modeling of public reasoning, and structured evaluations that align with the complexity of real-world decisions.

**Implications of Adopting Reasoning Fidelity as a Core Standard.** Adopting reasoning fidelity as a core standard would shift generative agent research from stylistic fluency to structural interpretability. It reshapes alignment evaluation, promotes modular and revisable architectures, and incentivizes cognitively grounded benchmarks. In high-stakes applications such as civic simulation, participatory policy, and AI governance, agents with causal transparency and revisable beliefs are essential for trust, auditability, and fairness. Without this shift, we risk institutionalizing brittle models that obscure bias and flatten the diversity of public thought.

**Open Challenges and Next Steps** We are actively developing:
- Agent architectures for modular belief reasoning and counterfactual revision;
- Tools for causal motif extraction and belief graph construction;
- Datasets across domains such as housing, surveillance, and healthcare.

Alongside these efforts, we identify several open challenges:
- Constructing causal belief networks from natural language transcripts remains challenging, due to ambiguity in concept identification, causal direction, polarity, and conceptual granularity;
- Causality alone cannot capture the full range of human reasoning. People also rely on associative, analogical, and emotional processes that resist strict symbolic modeling. Our initial focus on casuality is a strategic and computationally tractable starting point, not an endpoint.

We invite the community to co-develop evaluation protocols, agent designs, and data pipelines that advance cognitively aligned simulation.

**To simulate society faithfully, we must simulate thought.**

## Acknowledgments

This work builds on research originally developed at MIT. We thank Kent Larson, Jinhua Zhao, Paul Liang, and Deb Roy for their valuable guidance and discussions throughout this work. We are especially grateful to Kent Larson for his continuous mentorship, to Deb Roy for his conceptual insights and high standards that shaped the framing of this work, to Jinhua Zhao for his thoughtful resonance and encouragement, and to Paul Liang for providing key technical perspectives.

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
