# OpenReview forum: "Simulating Society Requires Simulating Thought"
_NeurIPS.cc/2025/Position_Paper_Track — NeurIPS 2025 Position Paper Track_

### Official Review · Reviewer_UDbt · 2025-08-08

**Significance:** 3
**Presentation:** 3
**Rating:** 7
**Confidence:** 3

**Summary:**

This position paper argues that the recent development of simulating a society with LLMs requires the simulation of thought. That is, it must be grounded reasoning, structured, revisable and traceable.
In essence, this means that we must focus on structuring their reasoning, rather than relying on just their outputs.
The position argues for constructing reasoning graphs via semi-structured interviews that LLM agents should utilise in social simulations to mitigate issues regarding diversity, fidelity, and traceability.

**Strengths:**

This paper provides solid evidence and is well embedded in the (cognitive science) literature for the proposed requirements; in particular, the motivating and illustrative examples are helpful. The responses to the alternative views are well-grounded in literature and examples.

**Weaknesses:**

The key concern regarding this position is that some of this seems to be in the works already. Reasoning LLMs can self-correct by simulating their thinking for a given problem when they find contradictions. While the presented framework concretises beliefs, and the current reasoning LLMs are susceptible to the issues the paper mentions, it seems already quite effective.

The other concern is that using Q&A-style interviews to structure reasoning and simulate thought could ultimately still have issues like hallucinations. I feel this could have been addressed more strongly in the alternative view regarding traceability. It is possible that I don't fully appreciate the position this paper proposes. However, it seems that such reasoning graphs, as in my previous example, could still easily capture errors that the framework attempts to mitigate.
Still, I think this position could spark a debate that pushes for better reasoning models.

**Questions:**

How does this position fit with the current advancements in reasoning LLMs that already implement a kind of thought simulation?

**Alternative Position:**

Yes, and alternative positions are well-considered and addressed by the argument

**Author Identification:**

No.

**Context:**

3

**Discussion:**

4

**Ethics:**

["NO or VERY MINOR ethics concerns only"]

**Position:**

Yes, the paper argues for or against a position related to machine learning.

**Support:**

4

**Thoroughness:**

3

---

### Official Review · Reviewer_QURT · 2025-08-08

**Significance:** 4
**Presentation:** 4
**Rating:** 9
**Confidence:** 4

**Summary:**

This paper is on simulating society with large language models (LLMs). Rather than generating plausible behaviour, cognitively grounded reasoning is suggested that is structured, revisable, and traceable. Although these language models can be improved through prompt engineering, fine-tuning, and other techniques, they lack internal coherence and causal reasoning. Belief traceability also needs to be constructed such that how people reason, deliberate, or respond to interventions should also be taken into consideration. Hereby the authors present two contributions: 1) A Conceptual Modelling Paradigm, called Generative Minds, that support structured belief representations for agents. 2) RECAP (REconstructing CAusal Path), a benchmark that assesses reasoning fidelity through causal traceability, demographic grounding, and intervention scenarios. The authors posit that we must concentrate on simulating thought patterns, rather than focussing on predicting next tokens, the approach resorted to by LLMs. From real-world, semi-structured interviews, how people explain and revise their beliefs can be captured, leveraging belief graph networks. RECAP can be used as a replicable schema for reasoning evaluation, not as a static dataset only.

**Strengths:**

The paper is very well-written, well-structured, and clear. What they suggest have potential in principle and practice to be implemented. The authors exemplify what they posit and their arguments are solid. They argue their opinions clearly in detailed and comprehensible manner. All the arguments they suggest are supported with reasoning and evidence. The topic is very relevant and important to the NeurIPS community. What they posit would have a large societal impact on LLMs and AI, since the current language models mostly concentrate on generating fluent texts, lacking coherent belief systems. Only focussing on predicting the next token sensibly might not be called real intelligence as also covered by the authors. Leveraging conceptual belief models and updating them consistently may be what society really needs as an evolving "entity", also caring about the diversity and different ideologies of individual persons. Once we simulate real thinking processes as mentioned in the paper, we can contribute to humanity overall in a more effective and practical manner.

**Weaknesses:**

The paper only has several weaknesses as I stated below:

1) Maybe some more details on constructing alternative conceptual belief networks along with their implementation difficulties could be mentioned. For example, different cultures and domains might need varying conceptual belief networks; in this case human in the loop processes could be incorporated in addition to the semi-structured approaches handled by LLMs as well, and it might be a bit labourious, based on such different scenarios.
2) Apart from the use of the em-dash character frequently that is also stated to be widely produced by AI tools (e.g., ChatGPT) a lot, the following typos and format issues had better be addressed as well:

i) Page 1. Line 18: .In the field of -> . In the field of (A white space to be added after the dot.)
ii) Page 8. Line 382: Design Principles. -> Please, move it to the next page so that the list header aligns better with the items

**Questions:**

1) As mentioned in the weaknesses part, could we incorporate human in the loop processes per different culture, ideology, etc. in a more effective manner and create different belief models, for example, per country or nation / religion, etc.? For example, some concepts can connote different meanings and polarities for various cultures, religions, and ideologies. If implementing all of these be considered to be labourious (not only using LLMs for semi-structured data), what approaches can be followed?
2) Can we leverage some already-existing ontology or conceptual models, such as ConceptNet 5, to make the process easier by updating these a bit? There are some similar studies in the literature and on-going projects being conducted by mostly top-tier companies, such as OpenAI, Google, and Meta.

**Alternative Position:**

Yes, and alternative positions are well-considered and addressed by the argument

**Author Identification:**

Yes, multiple of the authors.

**Context:**

4

**Discussion:**

4

**Ethics:**

["NO or VERY MINOR ethics concerns only"]

**Position:**

Yes, the paper argues for or against a position related to machine learning.

**Support:**

4

**Thoroughness:**

5

---

### Official Review · Reviewer_X714 · 2025-08-28

**Significance:** 3
**Presentation:** 3
**Rating:** 6
**Confidence:** 4

**Summary:**

The paper introduces Generative Minds (GenMinds), a cognitively grounded framework for modeling structured and traceable reasoning in LLM-based agents. To evaluate such agents, the authors propose RECAP, a benchmark assessing causal traceability, demographic grounding, and intervention consistency—advancing social simulation beyond surface-level behavior to deeper cognitive fidelity.

**Strengths:**

1. Clear Problem Identification with Practical Relevance
The paper clearly identifies the limitations of current LLM-based social simulations—such as shallow reasoning, hallucinations, and lack of interpretability—and frames these as critical issues in real-world applications like policymaking and stakeholder modeling.

2. Innovative Shift from Behavioral Mimicry to Cognitive Modeling
It proposes a compelling conceptual shift from surface-level behavior mimicry to cognitively grounded reasoning using Theory of Mind and modular reasoning traces. This shows originality and strong theoretical grounding.

3. Emphasis on Interpretability and Causal Traceability
The argument for modeling reasoning fidelity—including causal, compositional, and revisable belief structures—addresses a major gap in the field, potentially improving both trust and utility in high-stakes decision-making scenarios.

**Weaknesses:**

1. Lack of Concrete Implementation or Empirical Evidence in Introduction
While the ideas are conceptually strong, the introduction doesn't clearly state whether these frameworks (e.g., modular reasoning motifs, symbolic-neural models) are implemented or just proposed. It risks remaining at a theoretical level without demonstrated feasibility.

2. Ambiguity Around Scalability and Practical Deployment
Although the proposed approach claims better efficiency and generalization, it’s unclear how well this symbolic-neural framework scales to large, real-world datasets or complex simulations involving many stakeholders.

3. Limited Discussion of Limitations or Potential Pitfalls
The introduction does not preemptively address potential challenges, such as how modular reasoning would handle ambiguity in natural language or conflicting stakeholder beliefs, which are common in social contexts.

**Questions:**

See 'Weakness'

**Alternative Position:**

Yes, and alternative positions are well-considered and named but not addressed

**Author Identification:**

No.

**Context:**

3

**Discussion:**

3

**Ethics:**

["NO or VERY MINOR ethics concerns only"]

**Position:**

No, the paper presents new research without clearly advocating a position.

**Support:**

3

**Thoroughness:**

4

---

### Note · Authors · 2025-09-04

**1-10 Additional Comments:**

Overall, I found the position paper track to be valuable and important. It creates a unique space for work that provokes discussion and broadens the community’s horizons. The reviews were constructive and respectful, and I appreciate the engagement from reviewers. At the same time, I think the track would benefit from clearer guidance to reviewers on how to evaluate conceptual contributions, so that expectations align more consistently with the goals of the track.

**1-11 Submit Again:**

Probably yes

**1-1 Submission Process:**

4

**1-2 Next Year:**

I would like to see the position paper track continue to grow, with even more visibility and recognition across the NeurIPS community. It provides a much-needed space for conceptual contributions that stimulate debate and reflection.

**1-3 Future Development:**

It would help to provide clearer reviewer guidelines tailored to position papers, so that reviews focus more on conceptual originality, framing, and impact, rather than expecting full empirical demonstrations. At the same time, fostering discussions between reviewers and authors could enrich the feedback process.

**1-4 Interest:**

["Panel discussions with other position paper authors", "Structured debates on controversial topics", "Workshops for developing position papers", "Mentorship programs for early-career researchers"]

**1-5 Thoughtful:**

8

**1-6 Supportive:**

9

**1-7 Technical Aspects Versus Position:**

4

**1-8 Gate Keeping:**

7

**1-9 Camera Ready Changes:**

We will clarify more explicitly that our contribution is conceptual, not empirical, and highlight the purpose of stimulating discussion, reflection, and future exploration. We will also refine examples and framing to make this distinction unmistakable.

**3-1 Review Response1:**

QURT

**3-2 Reaction To Review1:**

We sincerely thank the reviewer for the thoughtful feedback, which gave us a strong sense of being understood and pointed toward meaningful next steps.

We acknowledge two main challenges in constructing conceptual belief networks from grounded human input. First, building reliable networks from transcripts is non-trivial. Issues include confusion between causality and correlation (RelatedTo mistaken for causal links), ambiguity in polarity (Competition → innovation vs. Competition → stress), rigid priors overriding lived evidence, and inconsistent concept granularity (joy vs. happiness, transportation vs. bus). Commonsense graphs such as ConceptNet 5.5 (as suggested) provide scaffolding but cannot fully solve these issues, highlighting an open challenge for the community.

Second, applying networks across scenarios and cultures is difficult. Even with the same words, meanings vary (e.g., ambitious as admirable vs. selfish). A basic but reliable approach is human-in-the-loop auditing of chatbot transcripts, which enables manual adjustments. Though labor-intensive, it is one of the most effective ways to ensure alignment, especially early on. Prior work on interactive ML (Amershi et al. 2014; Holzinger 2016) shows iterative human guidance can improve fidelity in complex or sensitive domains.

At the same time, scalable automation is also needed. Options include LLMs as translators between context systems, terminology mapping frameworks (e.g., BabelNet, ConceptNet 5.5), and multilingual embedding spaces (e.g., LaBSE, LASER) that encode latent semantic alignments. We see these not as replacements but as complements, building on human-in-the-loop reliability to extend reasoning fidelity across cultures.

In sum, these are scaffolding steps, not final solutions. Our proposal is a starting point and invitation for the community to move beyond surface-level prediction toward reasoning grounded in real human thought. We will also correct all typos and formatting issues.

**3-3 Review Response2:**

UDbt

**3-4 Reaction To Review2:**

We thank the reviewer for their thoughtful engagement. We also appreciate the recognition that our position may help spark broader debate.

We would like to clarify our scope. Our goal is not to develop general-purpose reasoning agents, but to replicate individual human subjects faithfully in social simulations. In this sense, our contribution should be seen as a shift from reactive agents, which map inputs to outputs without modeling others’ minds (Wooldridge & Jennings, 1995), toward deliberative agents that simulate beliefs, goals, and perspectives to guide social action (Bratman, 1987; Epstein, 2006). While recent reasoning models have made progress, such as contradiction detection and self-correction that work for many AI tasks, the central issue here is whether they faithfully capture how humans actually reason, deliberate, and revise beliefs.

On cognitive authenticity, we argue that post-hoc rationalization is misaligned at two levels. First, rationales are detached from the model’s actual reasoning. Zhao et al. (2025, arXiv:2508.01191) report that CoT reasoning is often distribution-dependent and brittle, suggesting it is more superficial than it appears. Second, rationales are detached from human reasoning, which is anchored in structured belief states. Without this grounding, existing models fall short of cognitive authenticity.

On functional sufficiency, outputs that merely “work” may be acceptable in general tasks, but for social simulation, sufficiency requires traceable causal revision. Otherwise, agents risk misleading civic or policy contexts, and untraceable interventions cannot be reliably evaluated.

This is why we emphasize reasoning fidelity and traceability. Our aim is not to replace all reasoning models, but to argue that for social simulation, reactive mimicry is insufficient. What is needed are deliberative, cognitively grounded agents that remain faithful, auditable, and diverse in their reasoning processes.

**3-5 Review Response3:**

X714

**3-6 Reaction To Review3:**

We thank the reviewer for the thoughtful comments and the recognition of the originality and grounding of our work. We appreciate the reviewer’s recognition of the originality of our work, even though some concerns reflected more of a method-paper perspective.

We would like to clarify that this paper was submitted to the Position Paper track, and its main purpose is to propose a conceptual framework and stimulate discussion, rather than to present a fully implemented system. The reviewer’s concerns about feasibility, scalability, and practical deployment are valuable. These are indeed open challenges, and while our paper does not provide full solutions, we view them as natural extensions of the framework. In ongoing work, we have begun collecting real human data and building early prototypes, which we hope will further develop the ideas presented here.

In this sense, as a position paper, we view the concerns raised less as limitations of the present work than as natural directions for extending the concepts we propose. Our intention in this paper is not to provide full solutions, but to introduce a framework that can stimulate discussion, encourage reflection, and guide future solutions. We will make this clearer in the revision by emphasizing that the contribution is conceptual rather than empirical.

---

### Meta-Review · Area_Chair_BaDW · 2025-09-06

**Rating:** 7
**Confidence:** 4

**Strengths:**

1. The paper’s motivation is clear and compelling. It presents an innovative shift from surface-level behavioral mimicry to cognitively grounded reasoning, leveraging Theory of Mind and modular reasoning traces. This demonstrates both originality and strong theoretical grounding.

2. The arguments are well-supported with evidence and firmly rooted in cognitive science literature. The motivating and illustrative examples are effective, and the responses to alternative perspectives are well-reasoned and well-supported.

3. The paper makes two key contributions: (1) Generative Minds, a conceptual modeling paradigm for structured belief representations in agents, and (2) RECAP (REconstructing CAusal Path), a benchmark for evaluating reasoning fidelity through causal traceability, demographic grounding, and intervention scenarios.

**Weaknesses:**

1. Limited Discussion of Limitations and Potential Pitfalls. The introduction does not sufficiently address potential challenges, such as how modular reasoning would handle ambiguity in natural language or reconcile conflicting stakeholder beliefs—issues that are particularly common in social contexts. I agree with the weakness identified by reviewer X714. Although the authors have addressed this point, a thorough discussion of the limitations remains essential.
2. Need for More Details on Constructing Alternative Conceptual Belief Networks. The paper could benefit from a more detailed discussion of how to construct alternative conceptual belief networks and the associated implementation challenges. For instance, different cultures and domains may require distinct conceptual belief networks. In such cases, incorporating human-in-the-loop processes, in addition to the semi-structured approaches facilitated by LLMs, could be valuable. This point was raised by reviewer QURT, and while the authors acknowledged that it is indeed a challenging task, I believe a more detailed description or discussion should be included in the revision.

**Questions:**

Please take the reviewers’ questions into careful consideration, as addressing them could make your paper much more solid and impactful.

**Ethics:**

All reviewers agree that this paper has NO or VERY MINOR ethics concerns only.

**Thoroughness:**

4

---

### Decision · Program_Chairs · 2025-09-26

Accept